## Original Research Article

mtDNA; mitochondrial dynamics; social networks; emergence.

**Author for correspondence:**
I. G. Johnston,
E-mail: iain.johnston@uib.no

# Exchange on dynamic encounter networks allows plant mitochondria to collect complete sets of mitochondrial DNA products despite their incomplete genomes

Konstantinos Giannakis[1], Joanna M. Chustecki[2] and Iain G. Johnston[1,3]

[1]Department of Mathematics, University of Bergen, Bergen, Norway; [2]School of Biosciences, University of Birmingham, Birmingham, United Kingdom; [3]Computational Biology Unit, University of Bergen, Bergen, Norway

## Abstract

Mitochondria in plant cells usually contain less than a full copy of the mitochondrial DNA (mtDNA) genome. Here, we asked whether mitochondrial dynamics may allow individual mitochondria to 'collect' a full set of mtDNA-encoded gene products over time, by facilitating exchange between individuals akin to trade on a social network. We characterise the collective dynamics of mitochondria in *Arabidopsis* hypocotyl cells using a recent approach combining single-cell time-lapse microscopy, video analysis and network science. We use a quantitative model to predict the capacity for sharing genetic information and gene products through the networks of encounters between mitochondria. We find that biological encounter networks support the emergence of gene product sets over time more readily than a range of other possible network structures. Using results from combinatorics, we identify the network statistics that determine this propensity, and discuss how features of mitochondrial dynamics observed in biology facilitate the collection of mtDNA-encoded gene products.

## 1 Introduction

Mitochondria are vital bioenergetic organelles, present in the vast majority of eukaryotic cells. Across and within eukaryotic organisms, mitochondria display a diverse variety of forms and dynamics. In plant cells, mitochondria largely exist as discrete, independent organelles. Unlike metazoan and fungal mitochondria, they rarely form large physical networks (with some exceptions; Seguí-Simarro & Staehelin, 2009). Individual plant mitochondria are highly dynamic, moving rapidly through the cell both along the cytoskeleton and diffusively (Logan, 2006; Logan & Leaver, 2000).

This physical population has a coupled genetic structure. Plant mitochondria do not typically contain full copies of the mitochondrial DNA (mtDNA) genome (Johnston, 2019; Preuten et al., 2010; Takanashi et al., 2006). Instead, many mitochondria either contain mtDNA 'subgenomic' molecules—encoding a reduced subset of mtDNA genes—or no mtDNA at all. The question arises: how do plant mitochondria maintain their protein complements, without a complete local genome from which to express new proteins?

One possibility (Arimura, 2018; Arimura et al., 2004; Logan, 2006; Takanashi et al., 2006) is that exchanges of subsets of mtDNA, mRNA and proteins between individuals can, over time, lead to the emergence of complete sets of mtDNA products in individual mitochondria over time. By mtDNA products, we refer to those protein products, tRNAs and rRNAs that are encoded by the complete mitochondrial genome. Protein products are expressed via mRNA transcripts; other gene products are expressed more directly from the mtDNA. As an example of this exchange, picture a mitochondrial genome which can be partitioned into two regions, A and B. One mitochondrion initially possesses a subgenomic molecule containing only Region A of the genome. Another initially possesses only Region B. Each expresses the genes contained in its subgenomic region. Then the two mitochondria physically meet and exchange their subgenomic molecules. The first

mitochondrion can now express genes from B, and vice versa. In parallel, RNA and protein populations of the two mitochondria can mix upon fusion events, so that even in the absence of mtDNA exchange, each organelle expands its complement of bioenergetic machinery. Within the dynamic cellular population of mitochondria, transient colocalisations indeed occur, resembling 'kiss-and-run' events in bacterial populations (Chustecki et al., 2021b; El Zawily et al., 2014; Liu et al., 2009; Logan, 2010). Some of these colocalisations result in transient fusion between two mitochondria. When fusion occurs, mitochondria can exchange genetic and protein material: indeed, mixing occurs through the entire cellular population on a timescale of hours (Arimura et al., 2004).

Recent work has characterised the 'encounter networks' between mitochondria in plant cells, describing which mitochondria encounter which others over time (Chustecki, 2021a; 2021b). Here, mitochondria are nodes, with two nodes linked by an edge if the corresponding mitochondria have been recorded within a threshold distance. Chustecki *et al.* showed that these encounter networks have structures which have the potential to facilitate efficient exchange of content, while also allowing mitochondria to spread evenly through the cell (Chustecki 2021b), and to adapt in the face of challenges (Chustecki 2021a). Hence, mitochondrial dynamics have the potential to resolve a tension between competing cell priorities: even spacing of mitochondria (with metabolic and energetic advantages) and colocalisation of mitochondria (for beneficial exchange of contents). This behaviour is one example of the many types of inter-organelle interactions in the cell (Cohen et al., 2018; Picard & Sandi, 2021; Valm et al., 2017).

Such functional encounters are an example of emergence, where the behaviour of a collective of individuals is different from the sum of individual behaviours. There are two coupled instances of emergent behaviour in our system—physical and genetic. First, the encounter network of mitochondria emerges from their underlying physical dynamics in the cell (Williams & George, 2019). Second, through exchanges of mtDNA, RNA and/or proteins within this encounter network, a complete set of mtDNA products for each mitochondrion may emerge. That is, over time, each mitochondrion will be exposed to a set of information greater than its own mtDNA complement, allowing the accumulation of the full set of mtDNA-encoded gene products. We hypothesised that the exchange efficiency of encounter networks could allow a mechanism for plant mitochondria to address their maintenance problem. Specifically, if mitochondria can efficiently exchange genetic information, transcripts and/or proteins, then the 'effective genome' to which each mitochondrion is exposed over time can eventually grow to include the full set of mtDNA-encoded gene products. To investigate this hypothesis, we proceed by using network science and quantitative modelling of exchange processes to investigate the genetic behaviours that these encounter networks could potentially support.

## 2 Results

### 2.1 The emergence of full sets of mtDNA products on Arabidopsis encounter networks as a network science problem

We first sought to understand the process by which full sets of mtDNA-encoded gene products could potentially emerge from dynamic interchange of molecules in plant cells, using encounter networks characterised from hypocotyl cells in 7-day *Arabidopsis* seedlings (see Section 4). In previous work, we established an experimental and computation pipeline to characterise the 'social'

encounter networks of mitochondria (Chustecki 2021b). Here, nodes represent mitochondria, and an edge between two nodes means that those two mitochondria have colocalised within a physical threshold distance at least one time point during the experiment (Figure 1). Example networks from mitochondrial dynamics in *Arabidopsis* hypocotyl are shown in Figure 1. We will use these, and other experimentally characterised encounter networks, in the subsequent analysis. Results from independent single cells were generally very similar (Supplementary Figure S1).

Our fundamental biological problem is: how can individual mitochondria become exposed to the full set of mtDNA-encoded gene products, given that each may only carry a reduced mtDNA molecule? To address this, we consider how exchange of subgenomic mtDNA molecules and gene products between mitochondria can accomplish this goal, informed by the set of physical encounters characterised above. While a physical encounter does not necessarily imply fusion and exchange of mitochondrial content, it is a requisite for this exchange. We therefore consider how the sets of mtDNA products present in sets of individual mitochondria change as a growing proportion of encounters are interpreted as leading to exchanges. On the one hand, if no encounters lead to exchanges and organelles begin without full product sets, mtDNA product sets will never become complete. On the other hand, if every encounter leads to an exchange, full product sets may emerge readily.

We present the specific phrasing of this problem in the Supplementary Material, illustrated in Figure 2. Qualitatively, we ask how many gene products an individual mitochondrion is exposed to over time, as a function of the proportion of encounters between mitochondria that lead to exchange of mtDNA and/or transcripts and proteins. Two situations are of interest for modelling this question. First, if each mitochondrion begins with little or no genetic information or gene products, how long does it take to accumulate the full set? Second, if each mitochondrion begins with a full complement of gene products but little genetic information, what proportion of gene products are retained as they decay over time? The model closest to reality for any given biological cell will fall between these extremes. If the degradation rate of transcripts and proteins is high compared to that of mitochondrial encounters, so that most mitochondria do not have full transcript and product complements, the first picture characterises the timescale upon which an organelle will be exposed to every gene. If transcripts and proteins are relatively long-lived compared to the timescale of mitochondrial encounters, the second picture characterises how individual mitochondria can retain full gene product sets in the face of transcript and protein degradation.

Some further quantitative connection with biology is possible here. A comprehensive recent survey of the *Arabidopsis* RNA decay landscape (Sorenson et al., 2018) has shown that mt-encoded RNA molecules have a wide range of half-lives. Some decay rapidly on the timescale of 1–2 hr [comparable to subhour degradation of some mtDNA transcripts in several other species (Gagliardi et al., 2001; Kuhn et al., 2001) and nuclear-encoded transcripts in *Arabidopsis* which degrade with a median half-life of 3.8 hr (minimum 0.2 hr; Narsai et al., 2007)]. Others remain strikingly stable over many hours. Electron transport chain proteins and other mitochondrial machinery degrade with a half-life of several days in *Arabidopsis* (Li et al., 2017). While longer-lived, many of these molecules are embedded in the inner membrane, and so are not as straightforward to exchange as RNAs which are mobile within the matrix. In onion epidermis, content mixing through the mitochondrial population (i.e., at least one fusion event for every

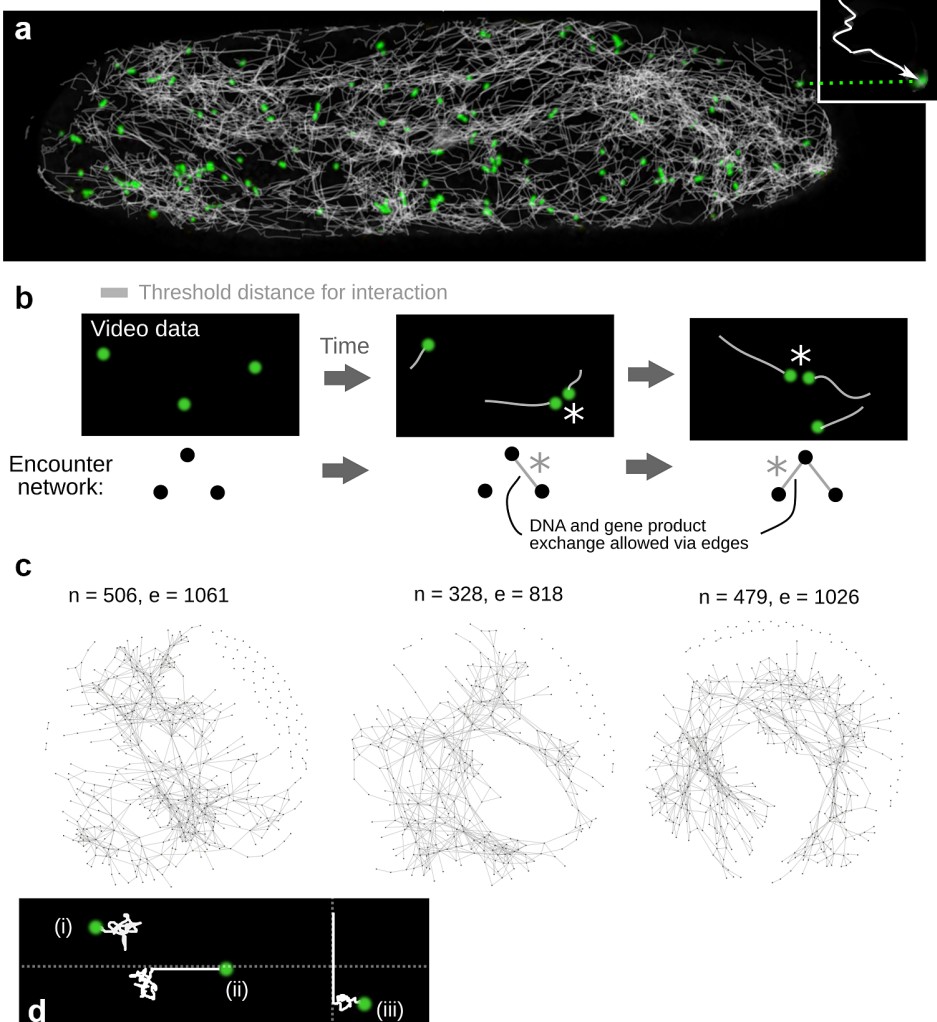

**Fig. 1.** Characterising mitochondrial encounter networks. (a) Confocal microscopy with mitochondrial green fluorescent protein (mtGFP) *Arabidopsis* (Logan & Leaver, 2000) creates videos of the motion of mitochondria (green) in hypocotyl cells. TrackMate (Tinevez et al., 2017) in Fiji (Schindelin et al., 2012) is used to characterise trajectories (white; individual shown in the inset). Individual mitochondria, as illustrated in the inset, may only carry a reduced mtDNA molecule encoding a subset of the full genome, along with a set of gene products. (b) Trajectory sets are interpreted as encounter networks by representing each mitochondrion as a node, and connecting two nodes with an edge if they are ever colocalised within a given threshold distance (∗). Encounters between mitochondria, if they lead to fusion and exchange, can expand the set of gene products in an individual mitochondrion. (c) Example encounter networks constructed over a period of 231 s: $n$ nodes and $e$ edges. (d) Simple physical model used to simulate mitochondrial motion. Model mitochondria may (i) move purely diffusively with constant $D$; (ii) attach to the cytoskeleton with probability $k_{on}$ per timestep and then move ballistically and (iii) detach from the cytoskeleton with probability $k_{off}$ per timestep and continue to diffuse. $k_{on}, k_{off}$ are parameters of the model which can take different values to capture different cytoskeletal influence (see Section 4).

mitochondrion) occurred on the timescale of 1–2 hr (Arimura et al., 2004). It therefore seems plausible that the exchange of molecules via mitochondrial encounters on the timescale of hours can support the emergence and maintenance of full sets of mtDNA-encoded products that degrade on a similar timescale.

Our network science questions share structural similarities with a wide range of problems in epidemiology (Akdere et al., 2006; Chakrabarti et al., 2008; Karp et al., 2000; Kempe et al., 2004; Moore & Newman, 2000), probability theory (including variants of the coupon collector problem; Flajolet et al., 1992; Newman, 1960), communication networks and algorithms (including the requirement for every node in the network to acquire required information about the existence of their neighbors; Vasudevan et al., 2009; Ye et al., 2012), but have some key differences (see Section 3). For brevity, we refer to these as *bingo problems*, by analogy with the collection of a set of elements which is built up

over time. A mitochondrion's *bingo score* is the proportion of gene products that it contains over a given threshold proportion $\epsilon$ of their maximum expression level. A *bingo* occurs when a mitochodrion has a bingo score of one, meaning that it contains the full set of mtDNA-encoded gene products at a level greater than $\epsilon$. An informative summary of a given cell's performance is the proportion $p$ of mitochondria that have scored bingos (the proportion of mitochondria that contain such a full set).

To characterise this behaviour, we simulated emergence via the 'bingo' games in Figure 2. We considered initial states that were either 'empty' (mitochondria begin only with one gene product, corresponding to their contained genetic element) or 'full' (mitochondria begin with all gene products—so every mitochondrion scores a bingo). In each case, we recorded the proportion $p$ of nodes that have scored a bingo (the proportion of mitochondria that contain all gene products) as a function of the proportion $q$ of

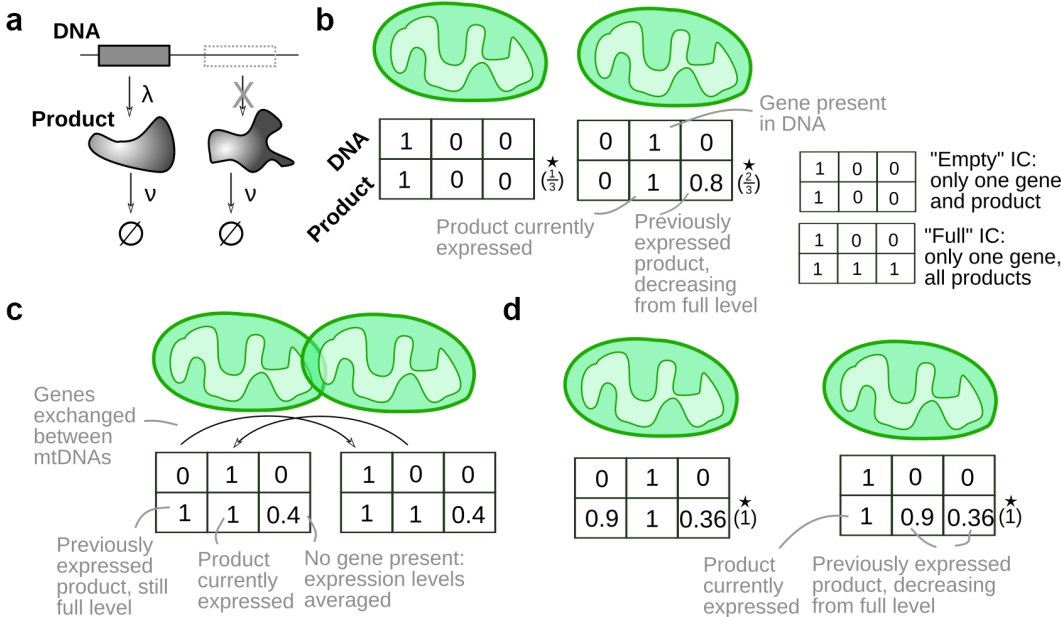

**Fig. 2.** Modelling emergence of full mtDNA product sets through exchange and complementation. Outline of the model for molecular exchange on encounter networks. (a) Effective gene expression model in this study. If a gene is present in an organelle, its product is produced with rate $\lambda$. Gene products degrade with rate $\nu$. We assume $\lambda \gg \nu$, so that product levels equilibrate rapidly when a gene is present. (b) Prior to an encounter, mitochondria have DNA and gene product complements. If a gene is present in an organelle, its product level is maximum 1, corresponding to active expression. In the absence of a gene, product levels may be nonzero due to an existing pool of transcripts or proteins. 'Bingo' scores—the proportion of gene products present at a level over $\epsilon$ (here $\epsilon = 0.1$)—are given under the stars. We consider two initial conditions (ICs): 'empty', where each mitochondrion only has products corresponding to its gene set; and 'full', where each mitochondrion has a full level of every product. (c) At an encounter, DNA sets are exchanged between organelles, and the levels of any products without a gene present are averaged. (d) Between encounters, product levels decrease unless the corresponding gene is present in that organelle.

encounters that correspond to an exchange. We increase $q$ following the temporal ordering of encounters in the network, and allow gene products to decay with a characteristic rate if their gene is not present in an organelle (Figure 2).

Intuitively, the dynamics of mtDNA product set accumulation depend strongly on $L$, the number of different genetic elements that are required to make up a full set (Figure 3a and c). For low $L = 2$, product sets rapidly emerge from "empty" initial conditions with low numbers of interactions, and in the $q = 1$ case where all edges lead to exchange, a majority of mitochondria are able to collect, or retain, a full set. For higher $L$, collection and retention become increasingly challenging, with, for example, only around 10% of mitochondria collecting a full set with $q = 1$ and $L = 5$, and fewer for higher $L$.

### 2.2 Arabidopsis encounter networks support efficient emergence compared to theoretical encounter networks

To assess the extent to which plant mitochondrial dynamics may be optimised for exchange of contents, we next asked how these biological networks compared to theoretical alternatives in their capacity to support such emergence. To this end, we investigated the bingo problem on a set of synthetic encounter networks.

For each experimentally characterised network, we built a range of synthetic networks constrained to have the same numbers of nodes and edges (Supplementary Figures S5 and S6). Our theoretical networks began with Erdős–Rényi (ER) random topologies (Erdős & Rényi, 1960; edges placed between pairs of nodes randomly chosen with uniform probability), scale-free (SF) topologies (Barabási & Albert, 1999; edges placed between pairs of nodes randomly chosen with probability proportional to their degree) and

Watts–Strogatz (WS) networks (Watts & Strogatz, 1998; Moore & Newman, 2000; a 'ring-like' network with subsequent rewiring to reduce network distances).

We further explored several other network types: geometric random graphs (GRGs) (Penrose, 2003), star graphs and 'cliquey' graphs. The final class followed our hypothesis that 'cliquiness' in networks would more directly lead to efficient genome emergence, as follows. Cliquey networks consist of cliques (sets of nodes that are all mutually connected) with few or no connections outside each clique. Nodes within cliques can then rapidly assimilate all available genes without risk of 'losing' them to a broader set of partners. We constructed two classes of cliquey network: (a) disconnected cliques of size $n$ and (b) cliques of size $n$ connected by a single link. In each of these synthetic cases, we specified a number of nodes to match a biologically observed network and padded the network with random edges if necessary to match that network's edge count.

We found that the bingo performance of different networks depends strongly on $L$, with some networks scoring higher than biological networks at $L \leq 3$ (ER, WS, geometric and small cliques) and lower at $L \geq 4$, and some with the opposite pattern (larger cliques; Figure 3b and d; Supplementary Figure S7).

This picture immediately suggests a tension in clique size. Smaller cliques will share information more rapidly. But if a clique is too small, it may not possess all the genes required to accumulate the full set. We found that for $L = 2$, bingo performance was a simple function of clique size, with smaller cliques (down to $n_c = 3$) performing best, and larger cliques (up to $n_c = 38$) performing worst. However, as $L$ increased, this picture became more nuanced. For $L = 3$, the performance of $n_c = 3$ networks was substantially challenged, due to the probability of a clique not possessing a

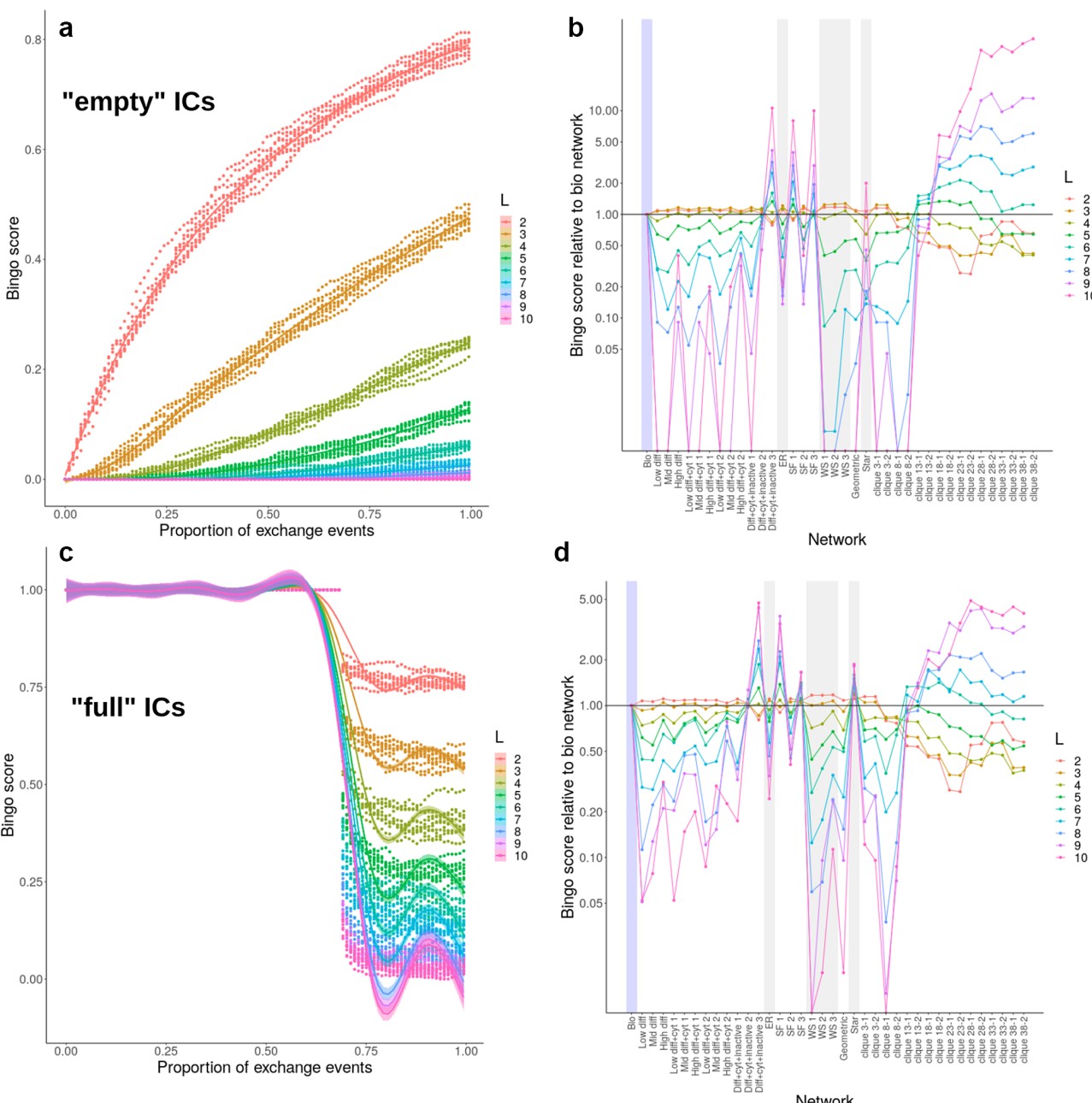

**Fig. 3.** Potential for mtDNA product set emergence on *Arabidopsis* encounter networks and on different theoretical network structures. (a,c) Empty initial conditions (ICs); (b–d) full ICs with gene product decay rate and threshold level $\nu = 0.02, \epsilon = 0.001$ respectively (other parameterisations behave similarly; Supplementary Figures S2–S4). (a,b) The 'bingo score' (proportion $p$ of mitochondria that have experienced a full mtDNA product set), as a function of the proportion $q$ of encounter network edges (physical encounters) that allow genetic exchanges. As $q$ increases, genetic information spreads through the mitochondrial population, and more individuals collect (or retain) the full set of genetic information, increasing $p$. This increase depends strongly on $L$, the number of different genetic elements that constitute the full product set: higher $L$ means more elements must be collected (or retained), which requires correspondingly more information exchange. Ten simulations were performed for each $L$ value, using an experimentally characterised *Arabidopsis* encounter network (see the text). (c,d) Final bingo score $p^*$, the proportion of mitochondria that possess a full product set if all encounters allow genetic exchange, is computed for all graph types. This plot shows this quantity normalised by the value for the biological network structure. For low $L$, some theoretical networks outperform biology but cliquey networks perform poorly. For high $L$, the situation is reversed. Traces connecting different network results are drawn to reflect the profile of results for a given $L$ and do not reflect any relationship between different networks. Networks immediately to the right of 'Bio' are encounter networks from physical simulation; others from synthetic construction. Labels: *diff*, diffusion; *cyt*, cytoskeletal motion; *inactive*, stochastic inactivation of mitochondria (modelling entering and leaving the domain); *ER*, Erdős–Rényi; *SF*, scale-free; *WS*, Watts–Strogatz; *clique x-y*, graph with cliques of size $x$, disconnected if $y = 1$ or connected by a single edge if $y = 2$. Different network classes appear on alternating grey and white backgrounds.

copy of each genetic element. For $L = 3$, larger cliques ($n_c = 8$) performed better, with even larger clique sizes ($n_c$ between 10 and 25) performing best for higher values of $L = 4$ to $L = 6$. Larger cliques $n_c > 30$ performed poorly in most cases, only becoming broadly competitive at high $L$ values.

However, the more striking result was that biological networks and synthetically constructed SF networks were the most robust performers. While never being the best performer for a given $L$, these networks performed much more consistently across a range of different $L$ values (Figure 3b and d). This suggests that

the mitochondrial dynamics generating biologically observed encounter networks provide a more robust way of collecting mtDNA products than a range of possible alternative dynamic and synthetic mechanisms.

## 2.3 Heterogeneous diffusive and ballistic motion supports efficient accumulation of mtDNA products

We next asked which properties of biological mitochondrial motion were responsible for the formation of encounter networks with strong bingo performance. To this end, we considered a simple physical simulation following Chustecki et al. (2021b) (Figure 1d and Section 4). Within the simulation, mitochondria move diffusively, with some probability of attaching to a cytoskeletal strand, whereupon they move ballistically until they detach with some probability. The attachment–detachment probabilities, diffusion constant and speed when attached to a strand are parameters of the simulation.

Exploring a range of parameters in this model (see Section 4), we found that no instance of the diffusive-ballistic model produced encounter networks that could outperform biological networks at bingo. While simulated performance was marginally higher for $L \leq 3$, performance at higher $L$ was substantially lower, only approaching the biological case for unphysically high values of the diffusion constant and ballistic speed (Figure 3b and d). The degree distributions (where degree is the number of connections between a node and its neighbours) of networks constructed through simulation typically had more limited spread, with fewer nodes of high degree (Supplementary Figure S6).

Pronounced inter-mitochondrial heterogeneity in dynamics has been previously reported (Chustecki 2021b; Logan & Leaver, 2000). Some mitochondria persist in a given cellular region for a long time period, whereas others enter and leave the region, leading to heterogeneity in the time windows for which a given mitochondrion is present. Those individuals present for longer have more opportunity to encounter partners and become highly connected. To model this, we introduced another process in our simulation model, allowing mitochondria to enter and exit the region of observation randomly with given rates (see Section 4). As before, we used simulations to produce encounter networks matching the node and edge count of the biological original. We found that these simulated networks, with high diffusion and cytoskeletal motion, more resembled the biological bingo performance (Figure 3b and d). Hence, a combination of diffusive and ballistic motion with broader variability in individual behaviour builds a foundation for efficient genome emergence.

To further explore this observation, we next artificially truncated the length of tracked trajectories in the biological data. Unsurprisingly, this led to smaller encounter networks, but also amplified the performance boost of SF and beneficially cliquey networks (Supplementary Figure S8). This observation supports the picture where a subset of individuals, remaining in the system for a comparatively long time period, accumulate more encounters and thus help facilitate the beneficial exchange of contents.

## 2.4 Network properties linked to efficient accumulation of mtDNA products

Given these observations, we next asked whether simple summary statistics of network structure correlated with bingo performance, and hence whether particular structural features might conceivably be selected in cellular control of mitochondrial encounter

networks. It may be anticipated that a network's performance at bingo would be related to how rapidly information can be spread through the network. This rapidity is captured by statistics like the global network efficiency $\nu = (n(n-1))^{-1} \sum_{i \neq j \in G} d(i,j)^{-1}$, the sum of the reciprocals of shortest path lengths $d(i,j)$ between all pairs of nodes $i$ and $j$, normalised by the number of pairs $n(n-1)$. Structural statistics like modularity (which we measure here using the walktrap algorithm; Pons & Latapy, 2006) and the size and structure of connected components may also be anticipated to play a role (the mean degree, by construction, is equal across all networks compared in an experiment).

However, when exploring bingo behaviour on our synthetic networks, we found that networks with high efficiency often do not perform well at bingo (Supplementary Figure S9). Other summary statistics also failed to show a tight correlation to bingo performance. While some correlated strongly for a given $L$ (e.g., increasing number of connected components decreases performance for $L = 2$), these relationships were typically reversed for different $L$ (increasing number of connected components increases performance for $L = 5$). One suggestive observation is that those networks that perform most consistently—SF and biological networks—have a high degree 'range', here defined as the number of values $k$ for which at least one node in the network has degree $k$ (Supplementary Figure S6). This quantity is at least somewhat related to the 'SF' nature of a network—degrees spanning a wide range of values— perhaps suggesting the capacity to accumulate information over a diverse ranges of 'scales' of $L$.

Given this observation, we next considered a more concrete theoretical framework to understand the problem of accumulating mtDNA products—specifically, the coupon collector's problem or CCP (Ferrante & Saltalamacchia, 2014). The informal phrasing of the problem is: if each cereal box contains a random coupon, and there are $n$ different types of coupon, how many cereal boxes do I need to buy to collect all $n$ types? Intuitively, this maps to the question of how many partners a mitochondrion needs to exchange with in order to collect all $L$ gene products in the system (Supplementary Figure S10). Given two results from the theory surrounding this problem, outlined and derived in the Supplementary Material, we are able to characterize and 'predict' the behaviour of a graph structure in bingo, including mitochondrial encounter networks, based on either individual node properties or more approximately using a simple scalar property of the network. Figure 4 confirms that the bingo game corresponds to the CCP solution described in equation (1) in the Supplementary Material. We see that the expressions from the CCP predict the game's outcome for the majority of network topologies and across different values of $L$ using either detailed information about individual mitochondria (Figure 4a) or a simpler summary statistic of the encounter network: the proportion of nodes with degree exceeding an $L$-dependent threshold derived in the Supplementary Material (Figure 4b).

These insights support the intuitive observation that nodes with degree less than $L$ can never score a bingo, and thus have a purely negative effect on the bingo performance of a network when measured by the proportion of bingo scores. Such nodes, including 'singletons' with degree zero, do occur in our biological encounter networks because of the limited time window of our observation (see Section 3). To check how much our general results depend on the presence of these low-degree nodes, we artificially removed degree-zero nodes from our biological encounter networks, and re-analysed these 'pruned' networks as above, constructing new synthetic and simulated networks to match the new node and edge counts. We confirmed that networks with 'pruned' and original

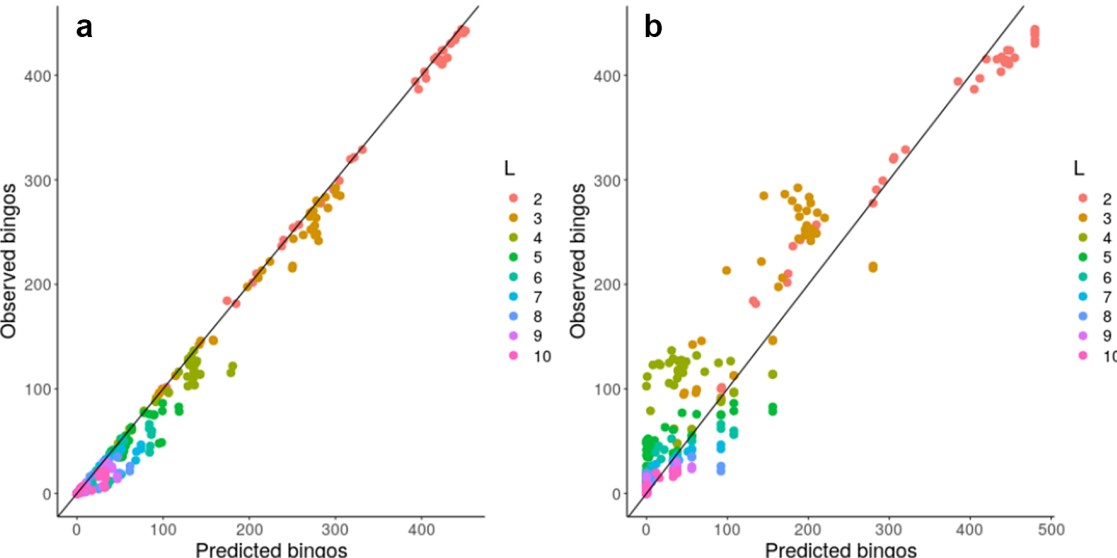

**Fig. 4.** Analytic results predict bingo performance on biological and synthetic encounter networks. Predicted (horizontal axis) and observed (vertical axis) number of nodes scoring bingo; each point corresponds to a different network. (a) expected results from equation (1) in the Supplementary Material; (b) estimated performance using the threshold value derived from the expected number of encounters needed to fill a bingo. There are 20 repetitions of each bingo game with different $L$, each value of $L$ is represented with a different colour.

statistics showed very comparable behaviours, demonstrating that the typically small proportion of singletons does not dramatically influence overall network performance at bingo (Supplementary Figure S8).

## 3 Discussion

The previous research presenting plant mitochondrial encounter networks (Chustecki, 2021a; 2021b) hypothesised that the collective dynamics of plant mitochondria allow the cell to balance two priorities. The first is an even physical distribution of mitochondria, ensuring a uniform energy supply, potential for colocalisation with other organelles throughout the cell, and avoiding heterogeneity in concentration of metabolites and signalling molecules. The second 'social' priority is colocalisation of mitochondria to facilitate exchange of genetic information and biomolecules. Here, we build on this second priority to show that the topology of encounter networks is capable of facilitating the efficient emergence of a complete mtDNA product set, through complementation of subgenomic mtDNA molecules and transcripts.

The main biological message of our model is that the observed dynamics of plant mitochondria theoretically support the collection of mtDNA product sets more efficiently than a wide collection of alternative, random and null models. A physical model capturing heterogeneous diffusive and ballistic motion of mitochondria mirrors the biological network behaviour, suggesting that this combination of random and cytoskeletal motion is the physical mechanism responsible, and required, for supporting efficient collection of mtDNA products through exchange. This in turn suggests several hypotheses. First, this dynamic behaviour may have been optimised by selection. Organisms with different mitochondrial dynamics (e.g., different balances of diffusion and cytoskeletal motion, as in the less efficient parameterisations of our physical model) would have lower capacities to collect mtDNA products by exchange, and thus may experience a selective disadvantage as mitochondrial protein complements are depleted. If this is the case, then we would expect to observe similar physical dynamics, giving encounter

networks with heavy-tailed degree distributions, in other plants too. Second, perturbations to this dynamic behaviour may compromise mitochondrial populations of mtDNA products. Here, we would expect to observe reduced levels of mtDNA products (even in bulk samples) of mutants compromising mitochondrial exchange, as DNA and transcript exchange become less able to compensate for product turnover. We would expect this effect to be particularly pronounced for genes with short transcript lifetimes. Testing of these hypotheses would be eminently possible through imaging mitochondria in other tissues and species (Chustecki et al., 2021b) and via proteomics in, for example, the FRIENDLY mutant compromising mitochondrial distribution and fusion (where mitochondrial functionality is already known to be compromised; El Zawily et al., 2014).

Our model makes quantitative predictions about the influence of inter-mitochondrion exchange on mtDNA product levels, which can be produced with a given set of biological parameters describing decay and exchange rates. For example, we may begin by assuming that all mitochondria form at least one connection on the timescale of 1 hr (Arimura et al., 2004). Consider two gene products that both decay with a 1 hr half-life [following Sorenson et al. (2018)]. These products are initially assumed to be present at a full expression level in every mitochondrion; each mitochondrion also contains a subgenomic molecule encoding one, but not both, of the products. In the absence of any exchange between mitochondria, after 2 hr, each mitochondrion will contain one product at its full expression level and the other at a 25% (and decreasing) level. If exchange is allowed, following our observed encounter networks, mitochondria will on average have a full level of one product and maintain a roughly 50% level of the other over time.

We have not considered mtDNA replication, degradation, recombination or other genetic dynamics (Johnston, 2019) in this model. Plant mtDNA readily recombines (unlike animal mtDNA), allowing mixing and restructuring of the information shared between mtDNA molecules (Broz et al., 2022; Woloszynska, 2009). Here, we only consider the question of mitochondrial access to gene products and genetic information, not the population dynamics

and/or restructuring of the molecules containing this information. This is a rich topic in itself, addressed by some classical (Albert et al., 1996; Atlan & Couvet, 1993) and some recent (Edwards et al., 2021) theories, and the influence of these physical dynamics of mitochondria on the genetic dynamics of mtDNA is an ongoing topic of research (Aryaman et al., 2019; Hoitzing et al., 2017; Johnston, 2019; Mogensen, 1996; Mouli et al., 2009; Poovathingal et al., 2009; Tam et al., 2013, 2015). We underline that the details of rates and magnitudes of our proposed mechanism remain hypothetical: although elegant experiments have demonstrated contents exchange and mixing throughout the chondriome (Arimura, 2018; Arimura et al., 2004), the physical and temporal scales of inter-organelle mtDNA exchanges remain, to the best of our knowledge, uncharacterised. Experimental characterisation of these processes will allow parameterisation of our model, which for now demonstrates the range of possible behaviours and general principles without specifying given parameter values.

Like any approach based on imaging, our characterisation of biological encounter networks is subject to some noise. The requirements to image the cell with a fine time resolution (so that mitochondria can be accurately tracked) and with limited laser power (to avoid damaging the cell) limit the resolution of individual frames, and the motion of mitochondria, while largely confined to a 2D plane, can sometimes lead to individuals being lost during the tracking process. This can affect the structure of the subsequent encounter networks. However, the most common issue—a mitochondrion being transiently 'lost' and hence, for example, being represented as two mitochondria (before and after the 'loss') early and late—will generally have the effect of reducing the degree of nodes. This is because the set of encounters of such a mitochondrion will be split between the two individuals. We thus expect the 'true' encounter network to involve more higher-degree nodes, thus supporting the distinction from the synthetic cases with limited degree distributions. On a similar note, our protocol involves imaging over a finite time window. Over time, encounter networks will gain more edges, and it is conceivable that over a long time the networks will come to resemble a complete graph, with every mitochondrion having encountered every other. However, there is another timescale in the system: the timescale on which genetic information is 'forgotten', as protein products expressed from a historically encountered genome molecule degrade. The system is thus expected to avoid steady-state behaviour, and our approach informs about the dynamics that shape the system in a sampled window of this out-of-equilibrium behaviour. Furthermore, plant cells are dynamic systems capable of responding to internal and external stimuli via sensing and feedback control. As such, the topology of a cell's encounter network is not fixed over the lifetime of the cell. Cells may adapt mitochondrial dynamics to favour, for example, 'cliquier' or sparser encounter networks as circumstances demand. The capacity of the cell to control mitochondrial dynamics to optimise mitochondrial exchange, and other priorities, is an exciting target for future work.

# 4 Methods

## 4.1 Plant growth

(Experimental protocols follow those in Chustecki et al. (2021b).) Seeds of *Arabidopsis thaliana* with mitochondrial-targeted green fluorescent protein (GFP) (kindly provided by Prof. David Logan; Logan & Leaver, 2000) were surface sterilised in 50% (v/v) household bleach solution for 4 min with continual inversion, rinsed three times with sterile water and plated onto $\frac{1}{2}$ Murashige and Skoog agar. Plated seeds were stratified in the dark for 2 days at 4°C. Seedlings were grown in 16-hr light/8-hr dark at 21°C for 4–5 days before use.

## 4.2 Imaging

Prior to mounting, cell walls were stained with 10 $\mu$M propidium iodide (PI) solution for 3 min. Following a protocol modified from Whelan and Murcha (2015), full seedlings were mounted in water on microscope slides, with cover slip. Imaging of dynamic systems in living cells is a balance between spatial/temporal resolution and maintaining physiological conditions. To avoid undesirable perturbations to the system including physical and light stress and hypoxia, all imaging was done maintaining low laser intensities and within at most 10 min of mounting to minimise the effects of physical stress and hypoxia (Prof. Markus Schwarzländer, personal communication).

A Zeiss 710 laser scanning confocal microscope was used to capture time lapse images. To test robustness of the imaging protocol, a Zeiss 900 with AiryScan 2 detector was also used for several identically prepared samples, with no differences between summary statistics collected from these samples and those from the 710 beyond natural variability. For cellular characterisation, we used an excitation wavelength of 543 nm and detection range of 578–718 nm for both chlorophyll autofluorescence (peak emission of 679.5 nm) and for PI (peak emission of 648 nm). For mitochondrial capture, we used an excitation wavelength of 488 nm and detection range of 494–578 nm for GFP (peak emission of 535.5 nm). Videos were 231 s long, with a frame interval of 1.94 s, and a resolution (after scaling for standardisation) of 0.2 $\mu$m per pixel.

## 4.3 Video analysis

Individual cells were cropped from the acquired video data using the cell wall PI signal using Fiji (ImageJ) (Schindelin et al., 2012). The size of each video was scaled to the universal length scale of 5.0 pixels/$\mu$m. We then extracted individual mitochondrial trajectories from the acquired video data using TrackMate (Tinevez et al., 2017). Typical settings used were application of the LoG Detector filter with a blob diameter of 1 $\mu$m and threshold of 2–7; filters were set on spot quality if deemed necessary. The Simple LAP Tracker was run with a linking max distance of 4 $\mu$m, gap-closing distance of 5 $\mu$m and gap-closing max frame gap of 2 frames. In each case, we visually confirmed that individual mitochondria were appropriately highlighted and that tracks were well captured, editing occasional tracks where necessary. XML output from Track-Mate was converted to adjacency matrices using custom code (see below).

## 4.4 Null model networks

We constructed several theoretical models for network structure, each with $n$ nodes and $e$ edges. First, ER random networks (Erdős & Rényi, 1960) were constructed by randomly choosing two non-identical nodes $a$ and $b$, each with probability $1/n$, and creating an edge between them, repeating until $e$ edges were created.

Second, SF networks (Barabási & Albert, 1999) were constructed by randomly choosing nodes with probability $1/deg(a_i) + 1/\sum_j 1/deg(a_j) + 1$. This procedure was repeated $e$ times, with degree updated each time, for the basic network (i). Variations of SF networks were created in two ways. For (ii), beginning

with a linear network where an edge connects each $a_i$ and $a_{i+1}$, then proceeding as in (i), thus enforcing connectivity. For (iii), a preferential attachment process was performed for each of the $n$ nodes, where a node is connected to a partner $a$ with probability $1/deg(a_i)+1/\sum_j 1/deg(a_j)+1$, where the sum $j$ is over nodes added so far to the network. Extra edges are then added as in (i).

Third, WS networks (Watts & Strogatz, 1998) were constructed as follows. Compute the mean degree $k = n/e$. Label each of $n$ nodes with successive integers. For each node $i$, draw $k_i = \lceil k \rceil$ or $\lfloor k \rfloor$ randomly with relative probabilities $\lceil k \rceil - k$ and $k - \lfloor k \rfloor$. If $k_i$ is even, connect $i$ to the $k_i/2$ nodes immediately before it and the $k_i/2$ nodes immediately after it in sequence. If $k_i$ is odd, connect to $(k_i + 1)/2$ 'before' nodes and $(k_i - 1)/2$ 'after' nodes with probability $\frac{1}{2}$, or vice versa with probability $\frac{1}{2}$. For all edges linking $i$ to a node with label $> i$, change the target node with probability $\beta$ to a different node $\neq i$.

Fourth, 'cliquey' networks were constructed. Given a clique size $c$ and constraints on $n$ and $e$, the number of cliques allowed was computed as $n_c = \min(\lfloor n/c \rfloor, \lfloor e/(c(c+1)/2) \rfloor)$. The $n$ nodes were partitioned into $n_c$ cliques with edges between each pair of nodes within each clique. These cliques were then either (a) left disconnected; (b) connected with a single edge linking two cliques; (c) left disconnected but padded with randomly placed edges to reach $e$ total and (d) connected with a single edge linking two cliques then padded.

Fifth, GRGs were constructed by placing $n$ points—each representing a node—in the unit square, and progressively adding edges between the two disconnected nodes with the shortest distance between their corresponding points, until $e$ edges existed. Finally, the star graph with $n$ nodes was constructed by connecting $n - 1$ nodes to a central node, then adding random edges until $e$ edges existed.

### 4.5 Model networks based on physical simulation

Synthetic encounter networks were constructed based on physical simulation of model mitochondrial dynamics using custom code in C (see below). As we are free to set length and time units in our simulation, we use $1\,\mu$m as the unit of length and set one discrete simulation timestep equivalent to 1 s. $n$ agents were simulated in a model cell, a 2D rectangular domain with reflecting boundary conditions at $x = 0$, $x = 100\,\mu$m, $y = 0$, $y = 30\,\mu$m, to model the geometry observed in our experimental observations of hypocotyl cells (Chustecki et al., 2021b). Cytoskeleton strands are modelled as crossing the cell at constant $x$ (horizontal) and at constant $y$ (vertical). Each agent could, at any time point, be detached or attached to the cytoskeleton. If detached, each timestep, agents were moved according to a normal kernel with standard deviation $2D\,\mu\text{m}^2$/s, so that $D\,\mu\text{m}^2$/s is the diffusion constant. When first attached, an agent is assigned a velocity vector: while attached, that agent moves by that vector each timestep. The velocity vector is randomly chosen on attachment and may be in the $+x$, $-x$, $+y$ or $-y$ direction, and has magnitude $V\,\mu$m/s. Each timestep, detached agents become attached with probability $k_{on}$, and attached agents become detached with probability $k_{off}$, corresponding to rates of $k_{on/off}$/s. When two agents were present within a distance $1.6\,\mu$m of each other, an edge corresponding to the pair was added to the encounter network (if not already present). The physical simulation proceeded until $e$ edges were present.

Characteristic values observed experimentally are $D \simeq 0.1\,\mu\text{m}^2$/s and $V \simeq 1\,\mu$m/s (Chustecki et al., 2021b). In our simulations, we explored one order of magnitude either side of these values, using $D = (0.02, 0.1, 1)\,\mu\text{m}^2$/s and $V = (0.1, 1, 10)\,\mu$m/s. We explored $(k_{on}, k_{off})$ pairs of $(0,0)$/s (no cytoskeletal motion), $(0.1, 0.1)$/s and $(0.5, 0.1)$/s.

Entry and exit of individual organelles into the system was modelled by switching individuals between 'active' and 'inactive' states. Active mitochondria behave as above and interact; inactive mitochondria remain static and do not contribute to any encounters, remaining effectively invisible (thus having exited the system). When this feature was used in simulations, activation and inactivation of individuals were stochastic events with rates $\rho_{on} = 0.01$/s and $\rho_{off} = 0.1$/s, respectively, leading to a mean of 10% active mitochondria at a given time.

## Acknowledgements

The authors are grateful to Morten Brun and Stein Andreas Bethuelsen for useful discussions, and to Storsåta for inspiration. We gratefully acknowledge the Imaging Suite (BALM) at the University of Birmingham for support of imaging experiments and thank Alessandro di Maio and Prof. Markus Schwarzländer for advice with design and analysis of imaging experiments.

**Financial support.** J.M.C. is supported by the BBSRC and University of Birmingham via the MIBTP doctoral training scheme (grant number BB/M01116X/1). This project has received funding from the European Research Council (ERC) under the European Union's Horizon 2020 research and innovation programme [grant agreement no. 805046 (EvoConBiO) to I.G.J.].

**Conflicts of Interest.** The authors have no conflicts of interest to declare.

**Authorship contributions.** I.G.J. conceived and designed the study. J.M.C. acquired experimental data. I.G.J. and K.G. developed code and analysis. I.G.J. drafted the manuscript; all authors edited the manuscript.

**Data availability statement.** All source code is available at github.com/StochasticBiology/mito-network-sharing. We used TrackMate (Tinevez et al., 2017) in Fiji (Schindelin et al., 2012) for video analysis, C with the igraph library (Csardi & Nepusz, 2006) for network generation and simulation and R (R Core Team, 2017) with libraries igraph (Csardi & Nepusz, 2006), brainGraph (Watson, 2020), XML (Temple Lang, 2020), ggplot2 (Wickham, 2016), GGally (Schloerke et al., 2021) and gridExtra (Auguie, 2017) for data curation and visualisation.

**Supplementary Materials.** To view supplementary material for this article, please visit http://doi.org/10.1017/qpb.2022.15.

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
