## [Reviewer Report]

Dear Editors,

We respectfully submit our manuscript "Encounter networks from collective mitochondrial dynamics support the emergence of effective mtDNA genomes in plant cells" for consideration in QPB.

Plant mitochondria, vital for energetic and metabolic processes across the kingdom, face an unusual problem. Unlike in mammals, individual plant mitochondria typically contain less than a full copy of the mitochondrial DNA (mtDNA) genome. At any given moment, a mitochondrion is therefore likely to lack the genes encoding several important elements of protein machinery. How then can individual mitochondria function, without a full local copy of the genome present to allow expression of all mtDNA genes?

Here we analyse a possible physical solution to this genetic problem. It is known that plant mitochondria move, meet, and exchange contents. We analyse the capacity of mitochondria to accumulate a full "effective" mtDNA genome through exchange with partners. Hence, although a mitochondrion may only ever carry a subset of mtDNA genes, exchanging this subset with partners can, over time, allow an individual to "see" the full range of genes in the mtDNA genome.

To explore this hypothesis, we take advantage of a recent approach combining timelapse confocal microscopy to characterise mitochondrial dynamics in single plant cells with video analysis to characterise the networks of encounters between mitochondria. Tools from network science and simulation allow us to model the exchange of genetic contents on these networks of encounters, and to characterise how the emergence of effective genomes proceeds on biological and synthetic encounter networks. We use approaches from combinatorics, specifically the coupon collector's problem, to make analytical progress predicting the efficiency of this emergence, which we show is strikingly high on biological encounter networks compared to other possible structures.

We believe this analysis and manuscript will be of interest to plant cell biologists, mitochondrial biologists, complex systems scientists and network scientists, as well as to a more general audience. To our knowledge this is the first theoretical attempt to link the physical and genetic dynamics of plant mitochondria in this way and we anticipate that it will open an avenue for followup theory and experimental studies.

For transparency, I am an Associate Editor at QPB and am involved in this capacity in the Emergence call. George Bassel, also involved in this call, is a friend and collaborator, and while I would be delighted if he handled this paper (especially given his expertise in network science applied to plant biology), I understand if this constitutes an editorial conflict of interest.

Many thanks in your time considering our submission.

Yours faithfully,

Iain Johnston (on behalf of the authors)

---

## [Reviewer Report]

*Comments to Author*: The study by Giannakis et al. is a continuation of the path of the Johnston group to provide a theoretical framework of fundamental mitochondrial characteristics including their genetics and their physical dynamics in the cell. These efforts are based on concepts proposed by several experimentalist and conceptual thinkers such as J. Allen, W. Martin, D. Logan and S. Arimura, who have provided empirical and conceptual, but typically not quantitative, evidence for their frameworks. A more comprehensive mathematical framework that enables not only validation but also allows to derive useful testable predictions has been lacking. The seminal work of the Johnston lab has started to fill this void by a succession of manuscripts. This is clearly much needed and deserves accessible communication at a prominent level.

With the present manuscript I have struggled providing an evaluation to the standard that I would usually strive for. This is because I can mainly judge the biological angle of this work. Yet, this work sets a clear focus on the theoretical network analysis, which I can only appraise superficially, which may not be appropriate. For a theory paper the text and the figures are presented well; yet as a biologist I have been struggling to follow. I do not feel that my own limits should be a measure by any means; on the other hand, I wonder if the presentation is pitched to the correct audience. As it stands I regard it not appropriate for a biological journal, even one with a quantitative focus such as QPB. Instead it may be very well placed in a journal with a theoretical, (bio)physical focus.

I would like to emphasize that this is likely an exceptional, creative and important study, and that my main concern is the target audience. Currently I cannot see a mitochondrial biologist to derive conclusions that go beyond the confirmatory level and drive further experimentation, while the topology of the network realized in the biological system is likely to be of considerable interest to the field of network theory. This concern applies at least for the current presentation of the work and becomes evident in the Discussion. A key conclusion is that the topology of encounter networks is capable of facilitating the efficient emergence of an effective genome (line 254). This may be regarded as trivial from a biological perspective without more quantitative statements on what is meant by efficient or effective. Other key conclusions (lines 261, 264, 269, 275) focus on the network theory rather than the biology.

A different presentation with a focus on the biology may well address this issue, but would entail major changes to the manuscript. If the authors decide to go down this path I believe the following would improve the usefulness of the work:

- Clarification of the assumptions made; putting them into context with the actual biological constraints, as suggested by empirical observation (data that is not available from the literature may need to be collected for a meaningful account).

- Clarification of the biological consequences of the findings.

- Clarification of the predictive (rather than descriptive) implications of the findings.

- Formulation of clear and testable hypotheses arising from the model.

To achieve this, the authors would need to translate their findings back into the specific biological context. The modelling approach is based on several assumptions, constraints and simplifications that may or may not carry biological meaning. For instance, video data are just limited for technical reasons; in real life the time has no such limitation. Is there really a need for a full genome, given that RNA and protein have considerable half-lives? How would the findings change with changes in mtDNA content and stoichiometry? Individual mitochondria can contain multiple nucleoids or different size. ‘Master circles’ are a conceptual construct and likely an artefact of the sequencing techniques. The physical structures of plant mitochondrial genomes are more likely to be linear and branched. This may not affect the actual model but deserves explanation to avoid detachment of the models from the biological situation.

These aspects may not be relevant for the network analysis, but they seem important to derive useful predictions about the biology. Here experimental measurements to define the most critical parameters empirically would be extremely informative; following up hypotheses derived from the modelling by experiments has very high conceptual promise, but this is not what has been done here.

Added biological value would also come from clearer quantitative statements on the biological consequences such as ‘Under assumptions XYZ, it would take XYZ encounters and XYZ minutes for a mitochondrion on average to have collected a full mitochondrial genome.

---

## [Reviewer Report]

*Comments to Author*: To start with, I should make it clear that I am writing this review from the point of view of a biologist interested in mitochondria, rather than an expert in mathematical or computational models. The modeling in the manuscript is explained clearly and I can find no fault with the math, but as I am not an expert in this side of the work I cannot be certain that there are no errors or omissions. I can, however, discuss the assumptions underlying the model and make suggestions that I think would make this work more useful for mitochondrial biologists.

The major problem is succinctly mentioned in the Discussion, but as a 'feature' rather than a design flaw, which is how I view it: 'One particular feature of our plant system is that information cannot be duplicated (mtDNA molecules are assumed not to replicate over the timescale of these dynamics). Once a mitochondrion has been exposed to an element, it remembers that exposure, but can only pass on the information from that element if it possesses an mtDNA molecule including it – whereupon it loses that molecule.' This is so unrealistic as to make conclusions from the modeling largely irrelevant to real mitochondria. In real mitochondria, what matters for the function of the organelle is not its DNA content, but its content of gene products. The RNA transcribed from the genome (and the proteins translated from the RNA) are present in many copies, can be shared during mitochondrial encounters independently of the DNA that encoded them, and are diluted rather than lost during exchanges. In the terminology of the manuscript, it is H, not G, that is likely to be exchanged during a mitochondrial encounter (although rather shared than exchanged). I think this would lead to very different results in the modeling; 'bingo' should be much easier and faster to achieve. Indeed, the results may even be trivial with the simple model described here. A truly useful model would incorporate some notion of kinetics (encounter rates, gene product production and turnover rates) and some notion of gene product concentrations to be able to follow turnover and dilution.

In conclusion, although this is an interesting exercise, the assumptions underlying the model are not realistic, making the conclusions of uncertain biological relevance.

---

## [Reviewer Report]

*Comments to Author*: Encounter networks from collective mitochondrial dynamics support the emergence of effective mtDNA genomes in plant cells

by Konstantinos Giannakis et al.

The study submitted by Konstantinos Giannakis et al. addresses the question whether plant mitochondrial dynamics may allow individual mitochondria to ‘collect’ an effective full copy of the mtDNA genome over time, by facilitating exchange between individuals similar to the trade on a social network. Therefore the authors analyzed the dynamics of Arabidopsis mitochondrial DNA dynamics using single-cell time-lapse microscopy, video analysis, and network investigation based on graph theory. Their results points to biological encounter networks that support the emergence of effective full genomes over time more readily than a range of other network structures. In conclusion the authors discuss biological backgrounds that may be responsible for the emergence of full effective genomes.

To merit publication, the authors should address the following comments.

General comments:

The authors should restructure the whole manuscript to better follow the addressed objective. Therefore the manuscripts needs a major revision which should contain clear objective, linked with results and conclusion. Several network analysis components described in the results section needs to be moved to methods or supplements (e.g. CCP). Further the biological component should be in focus through the whole analysis and the supporting rather then the statistics analogy. The content needs to be written in a way that a wide range of plant biologists can follow the analysis pieline.

What are the exact research questions? Here is a collection found in the manuscript that are only poor addressed or answered without a direct relation. This confuses the reader of the article and leads to a misunderstanding of the objective aims.

Line 12: Giannakis et al. addresses the question whether plant mitochondrial dynamics may allow individual mitochondria to ‘collect’ an effective full copy of the mtDNA genome over time, by facilitating exchange between individuals similar to the trade on a social network.

Line 58: We hypothesized that the exchange efficiency of encounter networks could allow a mechanism for plant mitochondria to address their maintenance problem.

Line 78: The biological problem is: how can individual mitochondria become exposed to the full mtDNA genome, given that each may only carry a reduced molecule?

Line 81: Qualitatively, we ask how many genes an individual mitochondrion is exposed to over time, as a function of the proportion of encounters between mitochondria that lead to genetic exchange.

Line 148: We next asked which properties of biological mitochondrial motion were responsible for the formation of encounter networks with strong bingo performance.

Lines 197ff: …we next considered a more concrete theoretical framework to understand the problem of effective genome emergence – specifically, the coupon collector’s problem or CCP

What is the exact definition of an effective genome?

Line 79: “We will refer to an ‘effective genome’ as the set of genes that a mitochondria has been exposed to over time.” With or without exchanging?

Abstract

In the abstract the results description is very poor and not fully understandable. The mentioned biological features that facilitate the emergence of full effective genomes should be shortly described as well as the mentioned network statistic.

Just to said,’…we identified the network statistic that determine this propensity…’ is to general and meaningless.

Minor comments:

Visualization comments: remove ggplot gray background and use colorblind friendly colors.

Figure 1: C) connected components mentioned in figure subscription but missing in the figure.

Figure 1. D) How exactly is the definition of kon and koff

Figure 2. Phrasing box within the figure needs to be removed and placed in the method section.

Figure 3. A and B Labels are missing. Please remove weird background.

Figure 3 A x-label: What is meant by proportion of edges used? Not mentioned in figure caption.

Figure 3 B: explain in the figure capture what is the definition of P0*? Also the authors explain that traces does not reflect relations between network types, the line is VERY weird.

Figure 3: please consider revising the axis labels to be less confusing, summarize text for 3B.

Figure 4: reformat the figure caption, remove spaces and change left/right with A and B to keep consistence.

Figure S4: Don’t think that someone can decipher this collection of descriptive statistics in one figure.

Major comment:

Related to figure 2, explain what happens if two genomes meet again which exchange genetic components (the 1 already exists in the history).

The explanation of the behavior of how encounters leads to an exchange is to short. Further the putative supporting Figure 3A is very confusing in this context. The reader does not understand how the bingo score behaves regarding the likelihood of exchange after encountering. Further the reason why only 10 simulation were performed showed in figure 3A is not clear. With q equals one, how can a mitochondria of length 10 get a bingo score of one?

Network performance is often mentioned within the manuscript but its still not clear what exactly it is.

Within the manuscript the authors often quantify results as e.g. “relatively good” or “relatively bad” etc. but compared to what kind of reference?

For example lines 132, 140, 143, 156, 171, 186, 188.

Line 143: What if biological networks have in general a scale free topology? This topology is very common in biological systems.

Lines 149-150: The authors should consider to detached the physical simulation from figure 1 (1D) and used at the right place in the manuscript to keep consistence and reduce confusion.

Line 158: Consider of explaining what is the meaning of degree distribution in the context.

Line 160: Who is others?

Lines 185ff: What is meant by network with high efficiency? What are values of other intuitively desirable statistics?

Why?? It is in every node’s interest to be the only node connected to as many other sources of information as possible; efficient networks typically connect “everything to everything”?? Reference for this statement needed!

What are other summary statistics?

The analogy of bingo game is a good method to explain some statistical behavior and it can be kept at very beginning to introduce the methodology. But for the manuscript content consider replace this analogy with the biological terminology and context.

It is very unclear how the coupon collector’s problem or CCP is related to the biological context; It is well explained from the statistical and graph analysis view.

In Figure 4 caption 20 repetitions are mentioned. Why 20? Same question like in Figure 3 why 10 simulations? Looks like an arbitrary choice.

Lines 227ff: Again what is here the definition of comparable behavior and network performance?

Regarding the explanation in the discussion about what is not considered in the analysis (lines 278ff), the authors need to strongly clarify why their analysis contributes to uncovering the mitochondrial DNA dynamic in plants.

Lines 313: Replace our bingo problem with meaningful biological terminology.

The link why plant mitochondrial dynamics represents a “social network” is not clear at all. There is not a single analysis that show similarities between social network and the presented encounter network. The network structure described that relates to scale free topology is also present in lateral gene transfer -networks for example or in several gene similarities networks.

---

## [Reviewer Report]

*Comments to Author*: Your article has now been assessed by three reviewers. Please apologize for the delays. I share with the reviewers the opinion that your work “Encounter networks from collective mitochondrial dynamics support the emergence of effective mtDNA genomes in plant cells” addresses important questions concerning mitochondrial genome dynamics. However, all three reviewers, representing experimentalists and theoreticians, concur and raised several major issues. These range (and non limited to) from the assumptions underlying the model and the biological relevance of the outcomes/conclusions, up to how the work is structured and readability/usability for biologists. I would therefore consider publication of your article provided it undergoes a major and thorough revision tackling the issues raised.

---

## [Reviewer Report]

*Comments to Author*: The authors have constructively addressed the comments from all three reviewers thrugh targeted, but extensive additions and rearragements. My specific concerns about linking the theoretical considerations to relevant biology have been convincingly dealt with and I believe the article will now be more accessible to its readership providing the opportunity to stimulate follow up activity - both experimental and theoretical. Except for minor spelling issues that will be taken care of during the production process I have no further comments and would like to congratulate the authors on a thought-provoking piece of work.

---

## [Reviewer Report]

*Comments to Author*: The authors have taken care to respond to the reviewers' suggestions, and I think the new version is much more relatable for biologists. Happily the conclusions are largely unaltered by the changes to the model.

Minor comment:

The manuscript cites Sorenson et al 2018 as source for short (~30m minute) half-lives for mitochondrially-encoded tRNAs. But this paper specifically looked at mRNAs not tRNAs, and indeed tRNAs are virtually invisible to RNA-seq without specific pre-treatment of the RNA (as explained in Warren et al 2021). Most mitochondrial tRNAs are not listed in the supplementary tables of Sorenson et al and the few that are have low read counts despite tRNAs being generally extremely abundant. I think it's safe to assume that whatever numbers Sorenson et al attach to tRNA IDs do not correspond to the processed (and unsequencable) mature tRNAs but to precursor transcripts. These precursors will have much shorter half-lives than the mature tRNAs. Generally tRNA half-lives are measured in days, not hours or minutes, although I'm not aware of any measurements of plant mitochondrial tRNA stability. So I suggest the authors pick a different example for a short-half-life gene product.

---

## [Reviewer Report]

*Comments to Author*: Your revised manuscript has now been evaluated by the two of the three previously involved reviewers. I apologise for the delays. Thanks for your efforts preparing a new version which is indeed improved. You have tackled most of the scientific issues. I am pleased to accept the manuscript for publication provided that you tackle the minor comment of Rev#1.

---

## [Reviewer Report]

Dear Editors, thanks for this great news and for your time and the reviewers'. We have removed references to tRNA halflives in response to R1 and have fixed some typos and references throughout.

---

## [Reviewer Report]

*Comments to Author*: Thanks for the submission of the revised version. I'm glad to let you know that your article has been accepted for publication.